# Retrospective Analysis of a Cohort of Patients with Metastatic Bladder Cancer with Metastatic Sites Limited to the Pelvis and Retroperitoneum Treated at a Single Institution between 2009 and 2020

**DOI:** 10.3390/cancers15072069

**Published:** 2023-03-30

**Authors:** Alexandre Bertucci, Lysian Cartier, Armelle Rollet, Rania Boustany, Werner Hilgers

**Affiliations:** 1Department of Medical Oncology, Sainte Catherine Cancer Institute, 84918 Avignon, Francew.hilgers@isc84.org (W.H.); 2Medical Oncology Department, Institut Paoli-Calmettes, INSERM, CNRS, CRCM, Aix-Marseille University, 13005 Marseille, France; 3Department of Radiotherapy, Sainte Catherine Cancer Institute, 84918 Avignon, France

**Keywords:** bladder cancer, radiotherapy, radiochemotherapy, oligometatasis disease, multidisciplinary approaches

## Abstract

**Simple Summary:**

Chemoradiotherapy as a consolidation treatment after chemotherapy in bladder cancer with lymph node metastasis limited to the pelvis and retroperitoneum has not been established. We retrospectively identified 502 patients who were treated with first-line chemotherapy for BC in our center. Patients with chemoradiotherapy or radiotherapy with an equivalent radiation dose superior to 30 Gy represented the RTCT group, and all other patients were included in the control group. A total of 89 patients were included, 24 in the RTCT group and 65 in the CT group. Chemoradiotherapy improved both OS (*p* = 0.034) and PFS (*p* = 0.009): 26.3 months (95% IC 0.0–52.9) and 19.4 months (95% IC 5.0–33.7), respectively, in the RTCT group versus 17.2 months (95% IC 13.7–20.6) and 11.2 months (95% IC 8.6–13.8), respectively, in the CT group. Grade 3/4 toxicity was related to chemotherapy and to chemoradiotherapy at levels of 31% and 24%, respectively. For mBC with metastatic regional or retroperitoneal lymph nodes, chemoradiotherapy seems to confer benefits for both OS and PFS.

**Abstract:**

Bladder cancer (BC) presenting with pelvic and retroperitoneal lymph nodes presents a therapeutic challenge. The impact of chemoradiotherapy on pelvic and retroperitoneal lymph node metastasis as a consolidation treatment has not been established. Between 2009 and 2020, 502 patients who were treated with first-line chemotherapy for BC in our center, were retrospectively identified. Patients who received chemoradiotherapy or radiotherapy with an equivalent radiation dose superior to 30 Gy were included in the RTCT group, and other patients were included in the control group (CT group). We performed an analysis of progression-free survival (PFS) and overall survival (OS) for these two cohorts using the Kaplan–Meier method. A total of 89 patients were included, 24 in the RTCT group and 65 in the CT group. Chemoradiotherapy improved both OS (*p* = 0.034) and PFS (*p* = 0.009) in comparison with chemotherapy alone: 26.3 months (95% IC 0.0–52.9) and 19.4 months (95% IC 5.0–33.7), respectively, in the RTCT group versus 17.2 months (95% IC 13.7–20.6) and 11.2 months (95% IC 8.6–13.8), respectively, in the CT group. Grade 3/4 toxicity was related to chemotherapy and to chemoradiotherapy at levels of 31% and 24%, respectively. For mBC with metastatic regional or retroperitoneal lymph nodes, chemoradiotherapy seems to confer benefits for both OS and PFS.

## 1. Introduction

Bladder cancer (BC) is the 10th most common neoplasm in the world with approximately 549,000 new cases per years and the 13th most deadly with approximately 200,000 deaths per years. The incidence of BC is higher in men and in developed nations. Some risk factors have been identified such as tobacco use, which may be attributed to at least 50% of cases, older age, male gender, Caucasian race, personal/family history of bladder cancer, prior cyclophosphamide and radiation therapy, schistosomiasis infection in endemic countries, obesity or chemical exposure to aromatic amines [1,2]. 

Approximately 5% to 10% of patients with muscle-invasive bladder cancer (MIBC) present at diagnosis with metastatic disease and 25% of patients treated for pT3–4 or pN1 disease will ultimately relapse [3]. Metastatic bladder cancer (mBC) remains a therapeutic challenge; the median overall survival (OS) is 14 months and the objective response is 40 to 50% of patients with cisplatin–gemcitabine chemotherapy regimen, the standard of care [4]. 

The investigations of several drugs such as sunitinib, lapatinib, bevacizumab, vinflunine or paclitaxel in maintenance treatment following a cisplatin-based regimen have shown no benefits to overall survival [5,6,7,8,9]. Only recently have immune checkpoint inhibitors (ICI) shown significant benefits in the second-line treatment of mBC [10,11,12,13]. In addition, maintenance therapy with avelumab (JAVELIN bladder 100) or pembrolizumab significantly prolonged the median overall survival at 21.4 months or 22 months, respectively, for patients who did not have disease progression with first-line chemotherapy in mBC [14,15]. 

In localized MIBC, radiotherapy has been integrated as part of a multimodality bladder-preserving approach for patients unfit for cystectomy [16]. In the metastatic setting, the impact of consolidative chemoradiotherapy (CRT) for increased local control or survival after chemotherapy has not yet been established. Oligometastatic disease in BC remains unclear and several definitions are found in the literature [17,18,19]. 

The role of radiotherapy in the treatment of localized and metastatic BC is evolving. Tri-modality treatment combining maximal transurethral resection, followed by concomitant chemo-radiation therapy presents a bladder-sparing alternative to radical cystectomy [20]. A recent French retrospective study suggests a benefit of CRT for both the OS and progression-free survival (PFS) in patients who have not progressed after first-line chemotherapy and with as much as five residual metastases [21]. Here, we present the results of our retrospective analysis. The aim of this study is to explore the impact of CRT in local control, distant recurrence and survival for mBC presenting metastases to the pelvic and retroperitoneal lymph nodes.

## 2. Materials and Methods

Between January 2009 and December 2020, 502 patients treated in our center with first-line chemotherapy for BC were retrospectively identified. Among them, patients presenting metastatic sites limited to multiple regional pelvic and/or retroperitoneal lymph nodes were analyzed. Inclusion criteria were pathological evidence (urothelial or other histology) of carcinoma originating in the bladder, ureter or renal pelvis; confirmation on imaging (TEP or MRI) of multiple regional lymph node invasion in true pelvis or to the common iliac (cN2/cN3) or beyond the common iliac but limited to retroperitoneum (cM1a). Patients who received CRT or radiotherapy with an equivalent dose superior to 30 Gy were included in the radiotherapy group (RTCT group), patients receiving less than 30 Gy (palliative radiotherapy) or no radiotherapy were included in the control group (CT group). We used the Bajorin prognostic model (prognostic score in untreated mBC), which includes two factors, a Karnofsky performance score (KPS) less than 80% and presence of visceral metastasis, to define 3 distinct prognostic subgroups with 0, 1 or 2 risk factors. Since the selection criteria of our study did not allow for visceral or bone metastasis, only patients with 0 or 1 risk factor are present.

Regarding descriptive data, we used Wilcoxon rank sum test, Fisher’s exact test and Pearson’s Chi-squared test to search a significative difference in both groups. PFS and OS for these two cohorts were estimated using Kaplan–Meier method. For univariate analysis, we used a stratified cox model to estimate hazard ratios (HR) and associated 95% CI for adjusted OS and PFS; variables with *p*-value < 0.02 are included in the multivariate analysis. Statistical analyses were carried out by a biostatistician from PRECIS, the Clinical Study and Health Innovation Research Pole, created by “AESIO Sante” in Montpellier.

This study followed the Strengthening the Reporting of Observational Studies in Epidemiology (STROBE) reporting guideline. This study did not require informed consent from participants because of the retrospective nature. A letter of no objection explaining the information of this study was sent to each living patient and patients could express their opposition to collecting their data at any time of the study. A copy of the written no objection consent is available for review by the editor-in-chief of this journal.

## 3. Results

Among the 502 patients who received treatment for BC in our center from 2009 to 2020, a total of 89 patients were retrospectively included in our analysis, 24 in the RTCT group and 65 in the CT group. Figure 1 shows the flow chart. 

The median age was 70, men represented approximately 80% of the population and bladder origins represented 90% in each group. Fifty patients (61%) had de novo metastatic disease. Seventy-eight patients (88%) had a favorable prognostic and eleven (12%) had an intermediate prognosis. Eighty-one patients (91%) had a urothelial carcinoma, two patients had a squamous cell carcinoma, two patients had a neuroendocrine tumor and four patients had other histology; 13.5% of patients received prior BCG therapy and 13.5% received immunotherapy with anti-PD(L)-1 in subsequent lines. No statistically significant differences were observed between the two groups regarding performance status, de novo or secondary metastatic disease, median hemoglobin level and the use of immunotherapy. No patient received immunotherapy before radiation. In the RTCT group, two patients had relapsed after bladder radiotherapy, one patient had a ureteral recurrence and received ureteral radiotherapy at relapse and one patient had a pelvic recurrence and received pelvic radiotherapy at relapse. Patient characteristics are summarized in Table 1. The median time between chemotherapy and CRT or radiotherapy was 4 months. Only four patients received chemoradiotherapy as the first treatment and three of them had a progressive disease and one a stable disease, which required the establishment of chemotherapy. Node involvement was validated in a multidisciplinary consultation meeting (RCP) by an expert radiologist, and no histological confirmation was carried out. In the RTCT group, nineteen (79%) patients received chemoradiotherapy with 45 Gy in 25 fractions with a boost of more or less of 18 Gy in 10 fractions; concerning the five patients having radiotherapy alone, they benefited from a 60 Gy regimen in 30 fractions. The radiotherapy treatment was limited to the pelvis, delivering 45 Gy to the bilateral lymph nodes and the bladder in 25 sessions of 1.8 Gy, without an overdose on the initially invaded ADP(s), and hen additional doses up to 63 Gy on the bladder (i.e., 18 Gy in 10 sessions of 1.8 Gy). If the patient had a retroperitoneum involvement, radiotherapy was extended to this lymph node region at a dose of 45 Gy. One patient had an RT to an ilio-obturator lymph node persistent after chemotherapy; the radiotherapy delivered a dose of 60 Gy in 30 sessions of 2.0 Gy, without a dose on the bladder or the ADP(s) initially invaded on the affected ureter and on the bilateral pelvic lymph node areas. No treatment was performed using stereotactic body radiation therapy (SBRT). We were also interested in the evolution over time of practices within our institution. We arbitrarily defined three time periods, between 2009 and 2012, 2013 and 2016, and 2017 and 2020, and the number who received CRT or radiotherapy accounted for 10, 6 and 8 patients, respectively. All patients were followed until progression or clinical deterioration with scheduled visits at our institution for physical exam and CT scan of the thoracic, abdominal and pelvic regions with 50% of visits every 2 to 3 months, 25% every 3 to 4 month and 25% every 5 to 6 months.

The median OS and PFS for all patients were 19.7 months (95% CI 16.5–22.9) and 13.4 months (95% CI 11.5–15.2), respectively. OS in the RTCT group was 26.3 months versus 17.2 months in the CT group with an HR at 0.52 (95% CI, 0.28–0.96) and *p* = 0.034. PFS in the RTCT group was 19.4 months versus 11.2 months in the CT group with a hazard ratio (HR) at 0.46 (95% CI, 0.25–0.84) and *p* = 0.009. Figure 2 and Figure 3 illustrate the OS and PFS, respectively.

In the univariate analysis of PFS, no group outside the treatment group reached significance, and consequently, no multivariate analysis was realized. In the univariate analysis of OS, the PNN/lymphocyte ratio and hemoglobin level reached significance (*p* < 0.2). In the multivariate analysis only, the PNN/lymphocyte ratio was statistically significant with *p*-values under 0.010. The hemoglobin level did not reach statistical difference with an HR of 0.86 (95% CI, 0.73–1.01). Concerning chemotherapy regimens, 7.7% and 25% received the MVAC (methotrexate–vinblastine–doxorubicin–cisplatin) protocol, 52.3% and 37.5% received a cisplatin-based regimen, 32.3 and 29.2 received a carboplatin-based regimen, and 7.7% and 8.3% received other regimens including paclitaxel and/or gemcitabine without platinum in the CT group and the RTCT group, respectively. The best response to first-line chemotherapy was progressive disease for 35.4% and 12.5% of patients, stable disease for 18.5% and 8.3% of patients, partial response for 29.3% and 63.5% of patients and complete response for 16.9% and 16.7% of patients in the CT group and the RTCT group, respectively.

Grade 3/4 chemotherapy adverse events were observed in 28 patients in both groups (31%) with no significant difference. Cytopenia was the most frequent adverse event concerning 14 patients, asthenia occurred for 13 patients (14.6%), and vomiting, infection, esophagitis and acute renal failure rarely occurred. Grade 3/4 radiotherapy adverse events were observed in six patients (24%), radiation-induced cystitis for two patients, diarrhea for one patient, thrombopenia and peripheral neuropathy for the same patient, anorexia for one patient and reversible acute renal failure for one patient. In both groups, no toxic deaths were observed.

## 4. Discussion

Our study aims to evaluate the addition of consolidative chemoradiotherapy (CRT) in the management of subdiaphragmatic lymph node metastases from BC for patients after chemotherapy. We report in our study that CRT in this setting improves both PFS and OS significantly.

We have only a little knowledge about the impact of targeted treatment such as radiotherapy on the N2/N3 and the M1a bladder cancer. In recent years, several small and retrospectives studies have helped to build the knowledge on this topic without addressing the question directly. Aboudaram et al. demonstrated a significant benefit of the CRT for mBC without progression and with no more than five residual metastatic lesions following the first-line systemic therapy, with OS enhancement both in the standard analysis (HR = 0.47, *p* = 0.015) and in the 6-month landmark analysis (HR = 0.48, *p* = 0.026); and with PFS enhancement only in the standard analysis (HR = 0.49, *p* = 0.007) [21]. Seisen et al. showed an OS benefit from an intensive primary tumor treatment compared to the standard of care in the context of mBC, with 14.92 vs. 9.95 months, respectively, *p* < 0.001 [18]. Shah et al. found long-term responders with CRT for patients who had a partial response after first-line chemotherapy. Out of 22 patients, 8 had no recurrence at 6 years [19]. Franzese et al. explored stereotactic radiation for oligometastatic BC, and they found an OS at 25.6 months with the number of metastases (1 site versus ≥2 sites) as a predictive factor; only 35.4% of patients had lymph node metastases. The patients may have received one to three prior lines of chemotherapy, 60.7% of patients had a single metastasis and 36.1% of patients were in oligo-progressive disease [22]. Abe et al. showed contradictory results with no benefit in OS for CRT after first-line chemotherapy; in this study, 66% patients had only lymph node metastases. They fixed the minimal equivalent dose at 50 Gy to consider no palliative radiotherapy, and 80% of patients receiving radiotherapy were treated for a single metastasis [17].

There is supportive evidence from other tumors showing that CRT confers clinical benefit. For example, in non-small cell lung cancer, Gomez et al. showed that the stereotactic radiotherapy, of three lesions or less with or without maintenance chemotherapy, brought a gain in PFS of 14.2 months (95% CI, 7.4 to 23.1) against 4.4 months (95% CI, 2.2 to 8.3), respectively, with *p* = 0.022, if the patients responded after a first line of systemic treatment [23]. In metastatic colorectal cancer, Lee et al. showed an increase in the median time to change to next-line systemic therapy from 5.0 months to 9.5 months with radiotherapy for oligometastatic disease (OMD) that progressed or persisted after first-line treatment. The study included diseases with five metastatic lesions or fewer [24]. 

We noted that the definition of OMD was of great variability between each study; most of the time, the definitions used in these studies were based on consensus, as in esophageal cancer, which defines OMD by one affected organ with ≤3 lesions in liver but ≤2 in lung or brain. In this meta-analysis including 15 studies, the authors found a favorable impact on OS with local treatment such as metastasectomy or stereotactic radiotherapy with a pooled adjusted HR at 0.47 (95% CI: 0.30–0.74) [25]. Moreover, in certain cancers such as metastatic prostate cancer, the radiotherapy of the primary and/or OMD has already entered clinical practice [26,27,28]. Currently, as in many cancers, OMD is ill-defined in bladder cancer unlike prostate cancer. 

All of the studies tend to show OMD as an entity at the border between the localized disease and the disseminated disease, associated with a better prognosis than diffusely metastatic cancers and probably with a continuum of malignancy [29]. The biology of OMD is poorly known and diagnosis is usually imaging-based. Since 2020, a working group from the European Organization for Research and Treatment of Cancer (EORTC) and the European Society for Radiotherapy and Oncology (ESTRO) has been conducting a project to find and harmonize a definition of OMD for all cancers as part of the OligoCare prospective study. This group is also focused on defining the concepts of the oligo-recurrence, the oligo-progression and the oligo-persistence [30]. However, integrating the dissemination pathways of each cancer in OMD is a necessary precondition in understanding this entity and will ultimately allow the therapeutic strategy to be improved.

Most of these studies simultaneously examined visceral, bone and extra regional lymph node metastases, prioritizing the number of metastatic sites rather than their anatomic location. In our study, we decided to focus on pelvic and retroperitoneal lymph node involvement only. These lymph node areas represent the first relay in the metastatic dissemination. Our study is designed with the hypothesis that these localizations represent the first step in cancer dissemination and are also associated with a lower risk of micrometastases and, consequently, further metastases. A focal treatment, such as radiotherapy, should be able to provide a clinical benefit in this situation. The results of our study suggest that retroperitoneal lymph node progression represents an oligometastatic entity in the bladder cancer.

Our study presents several biases limiting the extrapolation of our results, including the retrospective nature of our work, which by its nature presents a selection bias. The use of the experience of a single center, and an expert in radiotherapy, therefore leads to a tendency to favor radiotherapy in the treatment regimen of patients. Due to the definition of the population, we only report data on metastatic BC with a favorable or intermediate prognosis. Another limitation of our results is represented by the size of the cohort: the small number of patients limited the extrapolation of results. To be confirmed, our exploratory results must be verified on a larger number of patients prospectively. 

Treatment decisions regarding chemotherapy and/or radiation therapy were made after collegial discussion (medical oncologists and radiation oncologists) at the institutional tumor board. These treatment strategies have evolved over time, under the influence of experience with consolidation radiotherapy in other tumor types, especially NSCLC [31], and despite the evolution of practitioners, we have been able to maintain a certain homogeneity in these indications for chemoradiotherapy and in the management of oligometastatic patients. The median time of 4 months between the start of chemotherapy and the start of CRT shows a rapid access to radiotherapy as well as significant coordination between medical oncologists and radiation oncologists in our institution. The selection of patients eligible for OMD treatment requires significant expertise in radiotherapy in view of the absence of clearly defined recommendations.

## 5. Conclusions

Our work allows us to build up the hypothesis that consolidative chemoradiotherapy following first-line platinum-based chemotherapy in patients with subdiaphragmatic lymph node metastases of BC is associated with a clinical benefit regarding progression-free and overall survival and should be further investigated. Prospective investigations on the impact of consolidative chemoradiotherapy associated concomitantly or even sequentially with the maintenance immunotherapy in oligometastatic disease are also required.

## Figures and Tables

**Figure 1 cancers-15-02069-f001:**
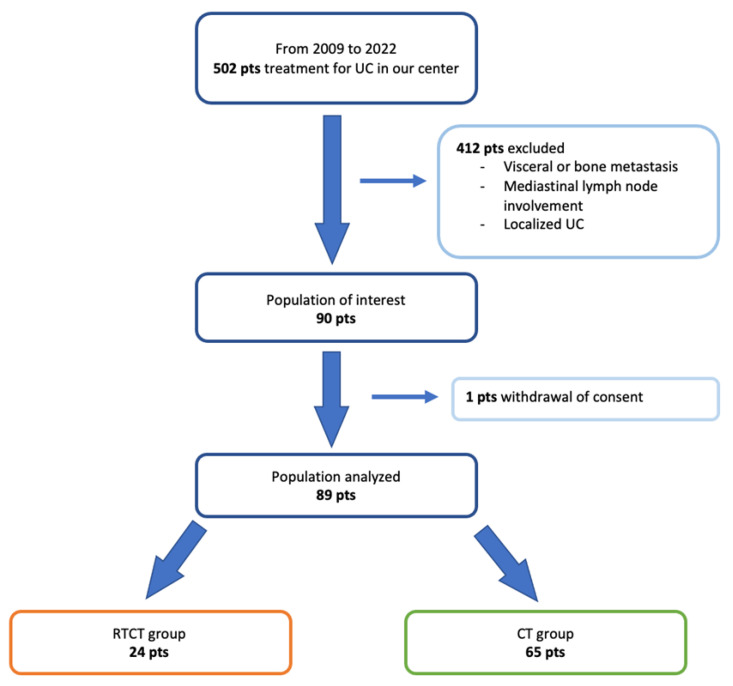
Flow chart. Pts: patients; UC/urothelial cancer; RTCT group: radiotherapy and chemotherapy group; CT group: chemotherapy group.

**Figure 2 cancers-15-02069-f002:**
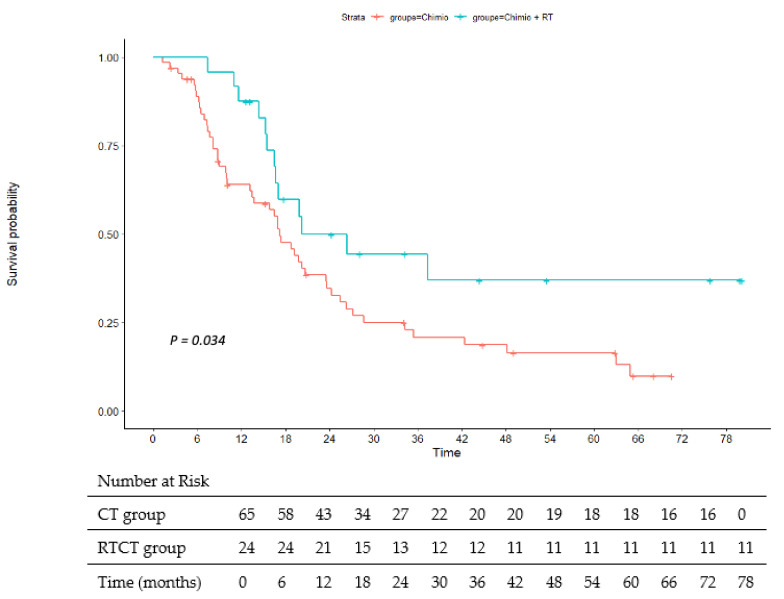
Overall survival. RT: radiochemotherapy; Chimio: chemotherapy.

**Figure 3 cancers-15-02069-f003:**
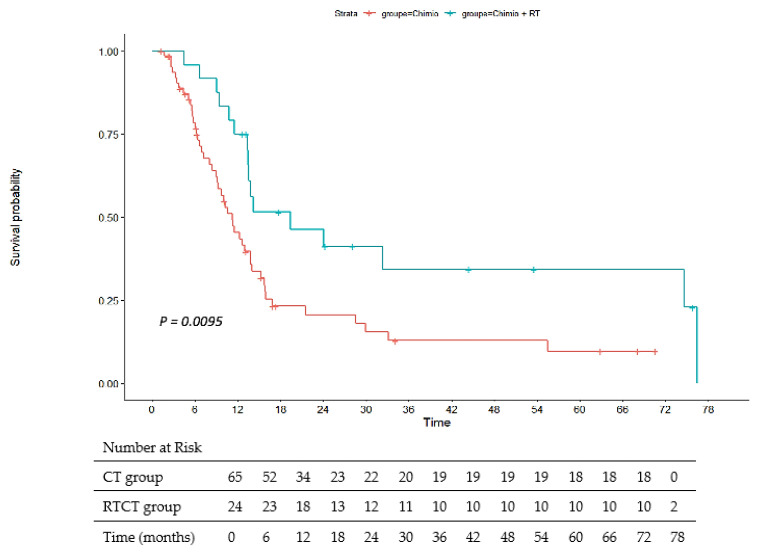
Progression-free survival. RT: radiochemotherapy; Chimio: chemotherapy.

**Table 1 cancers-15-02069-t001:** Demographic and clinical characteristics.

Characteristics	RTCT (no = 24)	CT Alone (no = 65)	*p*-Value ^1^
Median age (range)—yr.	69 (53–85)	70 (51–88)	>0.9
Age < 65 yr—no. (%)	8 (33%)	15 (23%)	
Sex—no. (%)			0.5
Male	19 (79)	55 (84.6%)	
Female	5 (21)	10 (15.4%)	
Performance Status—no. (%)			0.4
0	12 (50%)	29 (44.6%)	
1	11 (45%)	26 (40%)	
2	1 (4%)	9 (13.8%)	
3	0 (0%)	1 (1.6%)	
Histology—no. (%)			0.7
Urothelial	24 (100%)	57 (87.7%)	
NE	0 (0%)	2 (3.1%)	
Squamous	0 (0%)	2 (3.1%)	
Other	0 (0%)	4 (6.1%)	
Poorly differentiated—no. (%)	2 (8%)	6 (9%)	>0.9
Median Hb (range)—g/dL	13 (8.6–16.5)	11.8 (8.2–16.2)	0.2
No. of metastatic sites—no. (%)			
1	17 (71%)	38 (58.5%)	
2	7 (29%)	25 (38.5%)	
3	0 (0%)	2 (3%)	
Site—no. (%)			0.2
Pelvis	21 (88%)	50 (76.9%)	
Retroperitoneal	7 (29%)	33 (50.8%)	
Relapse after surgery	1 (4%)	7 (10.8%)	
Relapse after radiotherapy	2 (8%)	4 (6.2%)	
BCG therapy—no. (%)	3 (13%)	9 (13.8%)	>0.9
Immunotherapy—no. (%)	4 (17%)	8 (12.3%)	0.7
Metastatic first—no. (%)	16 (67%)	39 (60%)	0.6
Localization—no. (%)			0.7
Bladder	21 (88%)	59 (90.8%)	
Ureter	2 (8%)	6 (9.2)	
Renal pelvis	2 (8%)	3 (4.6)	

RTCT: radiochemotherapy; CT: chemotherapy; yr.: years; NE: neuroendocrine; Hb: hemoglobin; No: number. ^1^ Wilcoxon rank sum test; Fisher’s exact test; Pearson’s chi-squared test.

## Data Availability

The data presented in this study are available on request from the corresponding author. The data is not publicly available due to the no opposition letter provided to patients, which states that we will use the data for research purposes but will not disclose it.

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
