# Peer review of "Retrospective Analysis of a Cohort of Patients with Metastatic Bladder Cancer with Metastatic Sites Limited to the Pelvis and Retroperitoneum Treated at a Single Institution between 2009 and 2020"

_cancers, 2023, doi:10.3390/cancers15072069_

Round 1

Reviewer 1 Report

Congratulations on your well-written article that covers an extremely interesting topic. However, I would like to suggest some revisions that could improve the clarity and impact of your study.

Firstly, I believe that it would be beneficial to explicitly mention the small sample size as a limitation of the study. This would make it clear to readers that the conclusions drawn from the research may not be generalizable to larger populations.

Additionally, I would advise expanding the materials section by reporting more information on the type of follow-up performed before and after the diagnosis of the patients' metastasis. Finally, please consider adding the number at risk to the Kaplan-Meier curve figures

Author Response

Reviewer 1's Comments and Authors’ Responses:

Congratulations on your well-written article that covers an extremely interesting topic. However, I would like to suggest some revisions that could improve the clarity and impact of your study.

We thank the Reviewer for his/her positive comment.

Firstly, I believe that it would be beneficial to explicitly mention the small sample size as a limitation of the study. This would make it clear to readers that the conclusions drawn from the research may not be generalizable to larger populations.

We thank the Reviewer for his/her comment and we have modified the text by “Another limitation of our results is represented by the size of the cohort, the small number of patients limited the extrapolation of results. To be confirmed, our exploratory results must be verified on a larger number of patients prospectively.”

Additionally, I would advise expanding the materials section by reporting more information on the type of follow-up performed before and after the diagnosis of the patients' metastasis.

We thank the Reviewer for his/her comment and we have modified the text by “All patient were followed until progression or clinical deterioration with scheduled visits at our institution for physical exam and CT scan of thoracic, abdominal and pelvic regions  with 50% of visits every 2 to 3 months, 25% every 3 to 4 month, and 25% every 5 to 6 month.”

Finally, please consider adding the number at risk to the Kaplan-Meier curve figures.

We thank the Reviewer for his/her comment and we have modified figure 2 and figure 3.

Reviewer 2 Report

The authors present a retrospective cohort study evaluating the impact of radiation on patients with low-volume, metastatic urothelial carcinoma. This topic presents an area of interest and I commend the authors on their attempt to report their case experience. This work does suffer from significant issues related to selection bias and is retrospective in nature, and would benefit at least from an expansion on some of the available information to help determine generalizability and account for confounders.

Specific Comments/Questions:

It would be beneficial for the readership to see in table 1 if there are significant differences in at these variables between the two groups. For example, it appears that there is a difference in site of nodal disease.

Could the authors elaborate on the decision making process for adding radiation to these patients. Ie, has there been an institutional change over time that led to addition of radiation, a change in the patterns of a particular practitioner, or is this a reflection of selection. Having the time periods available for these patients would be of interest to frame the reader's thinking about the treatment landscape at the time.

It would be helpful to know more about chemotherapy regimens given, initial responses, and timing of chemotherapy. For example, time from chemo to radiation may be a significant confounder - patients who had a long interval from chemotherapy to radiation or those who had partial response to chemotherapy were likely to survive longer, and also may be the ones selected for consolidative radiation. There is no way to abstract this information from the provided data but would be presumably available to the authors. 

Can the authors clarify the radiation fields generally applied? Whole pelvis, SBRT to nodes, whole retroperitoneum, etc.

Did any patients have node biopsies for confirmation?

Author Response

Reviewer 2's Comments and Authors’ Responses:

The authors present a retrospective cohort study evaluating the impact of radiation on patients with low-volume, metastatic urothelial carcinoma. This topic presents an area of interest and I commend the authors on their attempt to report their case experience. This work does suffer from significant issues related to selection bias and is retrospective in nature, and would benefit at least from an expansion on some of the available information to help determine generalizability and account for confounders.

We thank the Reviewer for his/her positive comment.

Specific Comments/Questions:

It would be beneficial for the readership to see in table 1 if there are significant differences in at these variables between the two groups. For example, it appears that there is a difference in site of nodal disease.

We thank the Reviewer for his/her comment. We have modified table 1, we add the p-value.

Could the authors elaborate on the decision making process for adding radiation to these patients. Ie, has there been an institutional change over time that led to addition of radiation, a change in the patterns of a particular practitioner, or is this a reflection of selection. Having the time periods available for these patients would be of interest to frame the reader's thinking about the treatment landscape at the time.

We thank the Reviewer for his/her comment and we add “We are also interested in the evolution over time of practices within our institution. We have arbitrarily defined 3 time periods, between 2009-2012, 2013-2016 and 2017-2020, patients who received chemoradiotherapy accounted 10, 6 and 8 patients respectively.” in the result and “Treatment decisions regarding chemotherapy and/or radiation therapy were made after collegial discussion (medical oncologists and radiation oncologists) at the institutional tumor board. These treatment strategies have evolved over time, under the influence of experience with consolidation radiotherapy in other tumor types especially NSCLC [31] and despite the evolution of practitioners, we have been able to maintain a certain homogeneity in these indications for chemoradiotherapy and in the management of oligometastatic patients. The median time of 4 months between the start of chemotherapy and the start of CRT, shows a rapid access to radiotherapy as well as significant coordination between medical oncologists and radiation oncologists in our institution. The selection of patients eligible for OMD treatment requires significant expertise in radiotherapy in view of the absence of clearly defined recommendations.” in the discussion.

It would be helpful to know more about chemotherapy regimens given, initial responses, and timing of chemotherapy. For example, time from chemo to radiation may be a significant confounder - patients who had a long interval from chemotherapy to radiation or those who had partial response to chemotherapy were likely to survive longer, and also may be the ones selected for consolidative radiation. There is no way to abstract this information from the provided data but would be presumably available to the authors.

We thank the Reviewer for his/her comment, we have modified the main text, we add “Median time between chemotherapy and chemoradiotherapy was 4 months.” And we also add “Concerning chemotherapy regimens, 7.7% and 25% received MVAC (Methotrexate – Vinblastine – Doxorubicin – Cisplatin) protocol, 52.3% and 37.5% received cisplatin-based regimen, 32.3 and 29.2 received carboplatin-based regimen, 7.7% and 8.3% received other regimens including paclitaxel and/or gemcitabine without platinum in CT group and in RTCT group respectively. Best response to chemotherapy first line was progressive disease for 35.4% and 12.5% of patients, stable disease for 18.5 and 8.3% of patients, partial response for 29.3 and 63.5% of patients and complete response for 16.9 and 16.7% of patients in CT group and in RTCT group respectively.”

Can the authors clarify the radiation fields generally applied? Whole pelvis, SBRT to nodes, whole retroperitoneum, etc.

We thank the Reviewer for his/her comment and we add “The radiotherapy treatment was limited to the pelvis, delivering 45 Gy to the bilateral lymph nodes and the bladder in 25 sessions of 1.8 Gy, without overdose on the initially invaded ADP(s). Then additional dose up to 63 Gy on the bladder (i.e. 18 Gy in 10 sessions of 1.8 Gy). If patient had an retroperitoneum involvement, radiotherapy was extended to this lymph node region at a dose of 45 Gy. One patient had a RT to an ilio-obturator lymphnode persistent after chemotherapy, the radiotherapy delivered a dose of 60 Gy in 30 sessions of 2.0 Gy, without dose on the bladder or the ADP(s) initially invaded on the affected ureter and on the bilateral pelvic lymph node areas. No treatment performed in stereotactic body radiation therapy (SBRT)."

Did any patients have node biopsies for confirmation?

We thank the Reviewer for his/her comment, we have modified the main text, we add “Node involvement was validated in multidisciplinary consultation meeting (RCP) by an expert radiologist, no histological confirmation was carried out.”

Round 2

Reviewer 2 Report

I appreciate the revision provided by the authors as this gives clarity to important distinctions between groups.